# Alfred Russel Wallace's Intelligent Evolution and Natural Theology

**Michael A. Flannery**

UAB Libraries, University of Alabama at Birmingham, 1720 University Blvd., Birmingham, AL 35294, USA;
flannery@uab.edu

**Abstract:** Alfred Russel Wallace's conception of evolution and its relation to natural theology is examined. That conception is described as *intelligent evolution*—directed, detectably designed, and purposeful common descent. This essay extends discussion of the forces and influences behind Wallace's journey from the acknowledged co-discoverer of natural selection, to include his much lesser known position within the larger history of natural theology. It will do so by contextualizing it with an analysis of Darwin's metaphysical commitments identified as undogmatic atheism. In this sense, David Kohn's thesis that Darwin was the "last of the natural theologians" is revised to suggest that Wallace deserves to be included within the larger context of the British natural theologians in a surprisingly Paleyan tradition. As such, an important object of this essay is to clear away the historical fog that has surrounded this aspect of Wallace. That "fog" is composed of various formal historical fallacies that will be outlined in the penultimate section. Once described, explained, and corrected, Wallace becomes an enduring figure in carrying the British tradition of natural theology into the twentieth century and beyond.

**Keywords:** Alfred Russel Wallace; Charles Darwin; William Paley; evolution; natural theology

## 1. Introduction

My research into the life and career of the co-discoverer of natural selection, Alfred Russel Wallace, the so-called "other man" in the development of the modern theory of evolution, led me to one conclusion: "Wallace's understanding of the natural and metaphysical worlds eventually became one—an integrated whole of scientific, social, political, and metaphysical thought—forming a revised natural theology over the moribund special creation of William Paley" (Flannery 2018, p. ix).[1] While the details of that journey were evaluated in my *Nature's Prophet*, the context of Wallace's natural theology within the history of that vast and complicated enterprise is now explained.

If I could identify a missing piece to *Nature's Prophet*, it is that no summary analysis of natural theology's history was presented. Wallace's trajectory into that endeavor is simply given, with evidentiary support, as a brute fact. Here I would like to treat natural theology as it relates to Wallace with some degree of precision and detail. To begin, a durable definition of the term is demanded. Typically, natural theology is treated as a Christian apologetic, the attempt "to argue for the existence of God without resorting to special revelation" (Vainio 2017, p. 1), or variously as "the attempt to learn something of God from the exercise of reason and the inspection of the world from reflection on general experience rather than from specific revelatory events" (Polkinghorne 2006, p. 169). Both are

---

widely accepted definitions. The problem is that both definitions beg the question: Can a natural theology be constructed apart from this Christian-centric perspective?[2]

[1] Much of this essay is based on my book, *Nature's Prophet*, which discusses Alfred Russel Wallace's metaphysical ideas and its influences on and intersections with modern evolutionary biology (Flannery 2018). Readers interested in the original sources referenced in this essay are invited to consult the extensive notes found in that publication.

These two conventional definitions rule Wallace out of any consideration of natural theology; he was a non-Christian less interested in God as a Final Cause and more interested in the spiritual intersections between this physical realm and other forces or entities beyond this epistemic boundary. This being the case, a broader definition of natural theology is demanded. Here Alister McGrath offers a serviceable one as "the idea that there exists some link between the world we observe and another transcendent realm" (McGrath 2011, p. 12). With this as a working definition, Wallace becomes a good candidate to refurbish a natural theology more fully compatible with modern evolutionary theory. Here will be found strong connections with McGrath's definition along with some interesting surprises.

Wallace's project becomes clearer when it is fitted into the broader history of natural theology. The history of this ambitious effort can be found in two great periods. First, in the thirteenth and fourteenth centuries with the works of Bonaventure, Aquinas, and Duns Scotus (Swinburne 2007, p. 303); then again, with "its toadstool-like emergence in the seventeenth century," particularly in Britain, with a variety of figures such as Ralph Cudworth, Francis Lodwick, John Ray, and Nehemiah Grew (Gillespie 1987; Hunter 2001; Swinburne 2007). These distinct periods took two broad forms: The earlier natural theologians emphasized a priori reasoning and inferences of the order, harmony, and hierarchy of the universe to suggest a divine orchestrator—the "cosmic argument" (Gillespie 1987, p. 4); the latter group stressed the intricate design of nature to point to a designer—"physico-theology" (Gillespie 1987, p. 4; Polkinghorne 2006, pp. 170, 177). It is this later form that most influenced Charles Darwin and his colleague, Wallace, as they navigated through an intellectual heritage that had simply assumed the accuracy—in fact the *necessity*—of a natural philosophy premised upon a physico-theology propounded by sincere and zealous advocates, none of whom were more famous than eighteenth-century theologian William Paley.

Paley's principal works are his *Moral Philosophy* (1785), his *Evidences of Christianity* (1794), and his *Natural Theology* (1802). By the time Charles Darwin studied at Cambridge (1828–1831), Paley was that school's "most celebrated alumnus during the century after Newton" (Phipps 1983, p. 222). At that time, Darwin poured over *Moral Philosophy* and *Evidences of Christianity* with *Natural Theology*, giving him apparently "as much delight as did Euclid." Furthermore, although he felt these works were "of the least use to me in the education of my mind," he was "charmed and convinced by the long line of argumentation" (Darwin [1958] 2005, pp. 50–51). Despite Darwin's careful perusal of Paley's principal works, *Natural Theology* was never a set text at Cambridge, and theology had had "a relatively minor part of the formal curriculum, and natural theology played only a small role within that" (Fyfe 1997, p. 321). Given this context, Darwin's mention—even praise—of Paley, years later, seems all the more interesting.

The years following Darwin's graduation did little to diminish the importance of natural theology. The Bridgewater treatises initiated by the Earl of Bridgewater's will saw a concerted effort to modify and extend Paley's physico-theology into the next generation. While Darwin was examining the flora and fauna of the world on the HMS *Beagle*, Thomas Chalmers, John Kidd, William Whewell, Charles Bell, Peter Mark Roget, William Buckland, William Kirby, and William Prout attempted to establish convincing evidence to support St. Paul's declaration: "For since the creation of the world His invisible *attributes* are clearly seen, being understood by the things that are made, *even* His eternal power and Godhead, so that they are without excuse" (NKJV Rom. 1:20). Although technically not part of the Bridgewater treatises, Charles Babbage attempted to place an exclamation point to the project with his *Ninth Bridgewater Treatise* in 1837 (Brock 1966, pp. 175–76). The Bridgewater treatises captured the public's attention, receiving more than 120 reviews in over 40 publications (Topham 1998, p. 249). Read by the public less as theological and

---

2    Even a cursory review of history suggests that it can. Plato's Laws argued for a soul and Aristotle thought the cosmos gave evidence of a Prime Mover (Swinburne 2007, p. 303).

scientific works, they came to "represent a largely nontechnical, politically conservative, and religiously safe compendium of contemporary science" (Topham 1998, p. 241).

By mid-century, natural theology was alive but perhaps less than well. The Bridgewater authors had created an anthropomorphic God whose actions sometimes strained credulity and orthodoxy. The public was reading them but increasingly regarded these "Bilgewater" treatises as lacking in believability and soundness in answering the big questions of creation (Flannery 2018, p. 49). The answer came by way of Alfred Russel Wallace on the remote island of Ternate in the Malay Archipelago.

## 2. Wallace's Ternate Letter

Wallace had been in the Malay Archipelago (today known as the Southeast Asian Maritimes) for four years when he wrote the letter that would become "the central and controlling incident" of his life. It was written during a bout of malaria, probably while on the nearby island of Gilolo, and sent from the larger island of Ternate, the only island with regular British mail service in the region. The exact date it was sent can only be estimated because the original letter has not survived; a reasonable guess puts it at 28 March 1858 (Flannery 2018, p. 2). Received by Darwin on June 18, he immediately wrote to his close friend and confidant, Charles Lyell, "I never saw a more striking coincidence, if Wallace had my M.S. sketch written out in 1842 he could not have made a better short abstract! Even his terms now stand as Heads of my Chapters" (quoted in Flannery 2018, p. 2).

In one of the great coincidences of history, Wallace had independently discovered Darwin's theory of natural selection. This theory became the heart and soul of modern evolution, the idea that had prompted Darwin to exclaim, "I had at last a theory by which to work" (Darwin [1958] 2005, p. 99). Here was Wallace, an obscure collector of exotic specimens in far-off lands, a "specimen haggler" who sold his finds to the highest bidder back in London, presenting what appeared to be a duplicate of Darwin's precious theory. Like Darwin, Wallace used Thomas Malthus in presenting "a struggle for existence, in which the weakest and least perfectly organized must always succumb. Now it is clear," he added, "that what takes place among the individual of a species must also occur among the several allied species of a group." It is no wonder, then, that Darwin saw himself in Wallace's letter. Wallace's checks on population paralleled Darwin's action of selection, geological uniformitarianism, and environmental change. For Wallace, branching divergence suggested Darwin's branching tree, and they dug equally deep into geological time (Flannery 2018, pp. 32–33).

Their many similarities caused Darwin to panic. What should he do? Doing nothing might risk a preemptive publication by Wallace on the subject that Darwin had been laboring on quietly for years. Darwin's priority would be lost. Wallace had already done more publication on this topic than he had. Wallace's Sarawak Law paper, "On the Law Which Has Regulated the Introduction of New Species," had appeared in the *Magazine of Natural History* in its February 1855 issue and caught the attention of Charles Lyell, who had warned Darwin that Wallace was knocking on his evolutionary door, and Edward Blyth, curator of the museum of the Asiatic Society of Bengal in Calcutta, praised "Wallace's lucid collation of facts & phenomena" (quoted in Flannery 2018, p. 198 n. 7). Even worse, his Ternate Letter, "On the Tendency of Varieties to Depart Indefinitely from the Original Type" threatened to completely scoop Darwin.[3] After consulting with Lyell and his other close friend, Joseph Dalton Hooker, portions of Darwin's 1844 unpublished sketch (a 230-page elaboration on his earlier 1842 manuscript) along with an abstract of his letter to American botanist, Asa Gray, written on 5 September 1857, were chosen to be read at the next meeting of the Linnean Society. Accompanying these two presentations was Wallace's Ternate letter. Therefore, it was that on 1 July 1858, the modern theory of evolution, descent with modification by means of natural selection, was unveiled to the public.

---

[3] A portion of the Sarawak Law paper along with the complete Ternate letter with headings added later by Wallace in 1871 is now available in one source (Flannery 2018, pp. 167–87).

Fourteen months later, John Murray's London publishing house would make Darwin's magnum opus, *On the Origin of Species*, available to the general public.

If Darwin had read Wallace's letter more carefully, he might have noticed some significant differences of detail in their respective theories (Flannery 2018, pp. 33–42). One was Wallace's attention to group dynamics in demographic competition as opposed to Darwin's focus on species- specific, individualized competition. Another was Wallace's emphasis upon natural selection as a process of elimination versus Darwin's presentation of this principle as a creator of new forms. But perhaps their most important and sustained divergence came in their treatment of domestic breeding as an analogy for evolution in nature. Darwin made this his leading idea, and chapter one of his *Origin* was devoted to "Variation Under Domestication." Using examples from pigeon breeders to livestock, Darwin argued that humans could accomplish, by artificial selection in a comparatively short period of time, what nature could "select" given even greater time. Darwin even went so far as to anthropomorphize *natural* selection:

> It may be said that natural selection is daily and hourly scrutinizing, throughout the world, every variation, even the slightest; rejecting that which is bad, preserving and adding up all that is good; silently and insensibly working, whenever and wherever opportunity offers, at the improvement of each organic being in relation to its organic and inorganic conditions of life. (Darwin [1859] 1998, p. 84)

For Darwin, the analogy was not merely pedagogical, it was real and central to his theory. "In man's methodical selection," he insisted, "a breeder selects for some definite object, and free intercrossing will wholly stop his work. But when many men, without intending to alter the breed, have a nearly common standard of perfection, and all try to get and breed from the best animals, much improvement and modification surely but slowly follow from this unconscious process of selection, notwithstanding a large amount of crossing with inferior animals. Thus it will be in nature; for within a confined area, with some place in its polity not so perfectly occupied as might be, natural selection will always tend to preserve all the individuals varying in the right direction, though in different degrees, so as better to fill up the unoccupied place" (Darwin [1859] 1998, p. 102).

We will lay aside the question of how a "common standard of perfection" among breeders magically becomes "unconscious" in Darwin's scenario to ask a more pertinent question, how is the intentional breeding of these animals to approximate their own standards of "perfection" similar to nature? Wallace asked this same question and flatly rejected Darwin's analogy. His Ternate letter is clear: "in the domesticated animal all variations have an equal chance of continuance; and those which would decidedly render a wild animal unable to compete with its fellows and continue its existence are no disadvantage whatever in a state of domesticity. Our quickly fattening pigs, short-legged sheep, pouter pigeons, and poodle dogs could never have come into existence in a state of nature, because the very first step towards such inferior forms would have led to the rapid extinction of the race; still less could they now exist in competition with their wild allies" (quoted in Flannery 2018, p. 185).

In an interesting letter to Darwin on 2 July 1866, Wallace complained that otherwise intelligent people simply could not understand and appreciate "the self-acting and necessary effects" of natural selection. He believed this was almost entirely due to "your [Darwin's] choice of the term Natural Selection, and so constantly comparing it in its effects to man's selection, and also to your so frequently personifying nature as 'selecting,' as 'preferring' as 'seeking only the good of the species,' etc., etc." To correct misapprehensions as to the nature of the principle, he suggested "adopting [Herbert] Spencer's term (which he generally uses in preference to Natural Selection), viz. 'Survival of the Fittest.' This term is the plain expression of the fact; 'Natural Selection' is a metaphorical expression of it, and to a certain degree indirect and incorrect, since, even personifying Nature, she does not so much select special variations as exterminate the most unfavourable ones" (Marchant 1916, vol. 1, pp. 170–71). Here we see how Wallace clearly differentiated himself from Darwin in their conceptions of natural selection. Both would come to have a profound effect on the divergent trajectories of the theory they had both given birth to.

However, it is similarity, not difference, that holds our attention for now. In terms of natural theology, Darwin and Wallace both had a common target—the special creation made famous by William Paley. Nevertheless, their rejection of Paleyesque versions of nature would come to be expressed very differently, for reasons that become abundantly clear as this Darwin-Wallace story unfolds.

## 3. Darwin's "Barren Virgins"

As John O. Reiss points out, Wallace had fewer issues with teleology as he originally expressed it in his Ternate letter. This is because, in Reiss's words, Wallace has "a closer connection to the principle of the conditions for existence" (Reiss 2009, p. 145). What he means here is Wallace's reliance upon the principle of utility. This was stated by Darwin himself as follows: "Natural selection acts *solely* [emphasis added] through the preservation of variations in some way advantageous, which consequently endure" (Darwin [1859] 1998, p. 109). This was the core rationale of Darwin's theory. "I think it would be a most extraordinary fact if no variation ever had occurred useful to each being's own welfare," he wrote, "in the same way as so many variations have occurred useful to man. But if variations useful to any organic being do occur, assuredly individuals thus characterised will have the best chance of being preserved in the struggle for life; and from the strong principle of inheritance they will tend to produce offspring similarly characterised. This principle of preservation, I have called, for the sake of brevity, Natural Selection" (Darwin [1859] 1998, p. 127). As clear as this is, Darwin invited teleological difficulties by analogizing natural selection with artificial/human selection. This tension was only exacerbated by the fact that Darwin made chance, "spontaneous variation," and various expressions of fortuity central and indispensable to his overall theory (Flannery 2018, pp. 55–56, 61).

Darwin's materialism and rejection of design in nature ran deep, examples of which can be found in his early notebooks. For example, in Notebook C, during the spring of 1838 he wrote, "Thought (or desires more properly) being hereditary. —It is difficult to imagine it anything but structure of brain heredity, analogy points out to this—love of deity [is merely] the effect of organization. Oh you materialist!" (Darwin 1987, p. 291). Additionally, again in Notebook C, Darwin compares gradations in man (including mental gradations in races) to animals: "We see gradation to man's mind in a Vertebrate Kingdom in more instincts in rodents than in other animals & again in Man's mind, in different races, being unequally developed.—Is not [the] Elephant intellectually developed among Pachyderma like Man amongst Monkey—or dogs in Carnivora" (Darwin 1987, p. 299). He told Asa Gray on 22 May 1860, "I am inclined to look at everything as resulting from designed laws, with the details, whether good or bad, left to the working out of what we may call chance" (quoted in Flannery 2018, p. 61). For Darwin, the blindly constructive action of natural selection replaced the need for a designer. No wonder, he would write in his *Autobiography* that since the discovery of natural selection Paley's whole argument was destroyed (Darwin [1958] 2005, p. 73).

However, it was not just Paley that Darwin rejected. In his notes on John Macculloch's Proofs and Illustrations of the Attributes of God: From the Facts and Laws of the Physical Universe: Being the Foundation of Natural and Revealed Religion (1837), he declared "talk of Final causes" little more than "barren virgins" (Darwin 1987, p. 637). Darwin had little use for any of these design-in-nature arguments.

Despite his rejection of these Paleyan arguments, Darwin certainly engaged in his own brand of God-talk. This is important because it relates him to the larger context of natural theology itself and, therefore, helps to juxtapose his views with those of Wallace. Fact is, Darwin was constantly referencing God. Stephen Dilley believes that "Darwin utilized theology in order to help justify and inform descent with modification and to attack special creation" (Dilley 2012, p. 29). Dilley accepts Darwin's insistence that he was a theist when he wrote *Origin*. However, this comes from his *Autobiography* (Darwin [1958] 2005, p. 77), a troublesome source for the historian. Even though Darwin only intended this for his family (a bowdlerized edition was finally published for the public by his son, Francis, in 1893), Janet Browne, his leading biographer, has admitted that the *Autobiography* was a very self-conscious production. Darwin, "in choosing which memories to record in words, in selecting the

anecdotes [and beliefs?], he was constructing himself in the shape in which he wished to know himself and to be known by" (Browne 2002, p. 427). Darwin's claim of a theistic *Origin* cannot be taken at face value; passages from his notebooks might well suggest otherwise. Quoting the concluding passage in *Origin* that his theory of evolution "accords better with what we know of the laws impressed on matter by the Creator," Dilley concludes that Darwin "articulated a semi-deism in which God originally impressed the laws of nature onto matter and later directly created the first life, but then let unbroken natural law govern the unfolding of organic evolution" (Dilley 2012, p. 49). Of course, this leads to the awkward invitation for God to intervene at some other point in the natural history of the earth. Dilley then, more revealingly, writes:

> These were murky waters. Darwin's strategy, consciously or otherwise, was to avoid clarifying the matter. Vague God-talk had the disadvantage of blurring the true theological moorings and implications the *Origin*'s new science; it had the considerable advantage, however, of allowing deists and theists to interpret the *Origin* in the image of their own gods, making evolution by natural selection appear more persuasive. In this case, perhaps Darwin ignored theological clarity for the larger purpose of scientific success. (Dilley 2012, p. 49)

This is an essential fact to grasp in understanding Darwin. He was well aware of the controversial nature of his work. As one rhetorician put it, "*Origin* is anything but routine scientific rhetoric; Darwin writes nothing in *Origin* that does not have the audience in mind" (Moore 1997, p. 108). He believes "Darwin included theistic language for its rhetorical effect. Indeed, his use of words, phrases, and images whose theological connotations could not have escaped him is clear evidence of his concern for popular comprehension, even at the price of scientific precision" (Moore 1997, p. 112). Yet, how does this accord with Dilley's insistence that *Origin* represents a merger of science and theology in the form of semi-deism (Dilley 2012, p. 56)? It is probably closer to the truth to not see Darwin as a theist of any kind. Ultimately, Darwin claimed to be an agnostic, but even here he cannot be taken at his word. If Darwin thought that the laws were impressed on nature by a Creator, why did he rely upon the physical conditions of chance and necessity in thinking about the origin of life? He wrote to Hooker:

> It is often said that all the conditions for the first production of a living organism are now present, which could ever have been present.— But if (& oh what a big if) we could conceive in some warm little pond with all sorts of ammonia & phosphoric salts,—light, heat, electricity &c present, that a protein compound was chemically formed, ready to undergo still more complex changes, at the present day such matter wd be instantly devoured, or absorbed, which would not have been the case before living creatures were formed. (Darwin 1871)

Still, Darwin always kept a Creator close at hand. In referring to primordial origins he alluded to a seeming vitalism "into which life was first breathed" (Darwin [1859] 1998, p. 484). But he expressed private regrets to Hooker: "I have long regretted that I truckled to public opinion & used Pentateuchal term of creation, by which I really meant 'appeared' by some wholly unknown process" (Darwin 1863). This public rhetoric and private conviction can be reconciled if we get a handle on the simmering "agnosticism" that brewed within Darwin even as he wrote in his notebooks in the 1830s. Maurice Mandelbaum said it best:

> In the end his Agnosticism was not one brought about by an equal balance of arguments too abstruse for the human mind; it was an Agnosticism based on an incapacity to deny what there was no good reason for affirming. Thus, those who, at the time, regarded Agnosticism as merely an undogmatic form of atheism would, in my opinion, be correct in so characterizing Darwin's own personal position. (Mandelbaum 1958, p. 376)

Taken as a whole, this undogmatic atheism is confirmed in Darwin's notebooks and in numerous passages public and private. This is why Darwin could vehemently disagree with Asa Gray's theistic additions to evolution, and at the same time, use Gray to promote his theory. Darwin put Charles Kingsley to

similar use. As one analyst has said, "he [Darwin] was not above using others, like the Reverend Charles Kingsley, who cheerfully accepted his theory and thought God could simply be put on top of it. When Kingsley praised *Origin*, Darwin quickly put his words in the next edition, along with the sop about the Creator at the end. *He* didn't believe it, of course. The entire argument was designed to make God superfluous. But if folks like Kingsley thought they saw a place for God redundantly riding atop the whole thing, all the better for the ensuing propaganda campaign" (Wiker 2009, p. 99).

This undogmatic atheism allowed Darwin to use his God-talk with a freedom and plasticity unparalleled with the Paleyesque natural theologians. Cornelius G. Hunter believes that Darwin's theory was not antireligious, but instead depended upon a God of *his* creation. In Darwin's God-talk, he is constantly suggesting that no god worth our respect would have created a natural world with so much suffering. Darwin could, in effect, create a theodicy with a god refashioned in his own image. Yet, this is a form of atheism. As Hunter explains, "Darwin believed he had sufficient evidence to show that God would not have created this world. God's world had to fit into certain specific criteria that humans had devised" (Hunter 2001, p. 13). I can hardly think of a better definition of undogmatic atheism than this. The ubiquitous nature of this God-talk caused historian David Kohn to write about Darwin's "ambiguity" in his evolutionary theory. "The order implicit in the operation of natural processes did not rule God out," according to Kohn, "and could be so spoken of as to suggest a deity. This was no personal deity, no traditional Christian deity, no defined deity" (Kohn 1989, p. 238). It was *Darwin's deity*. But Kohn equivocates, writing, "When we learn to read nature, God is not there," and in this sense he calls Darwin "the last of the natural theologians" (Kohn 1989, p. 238). Again, we are back to Darwin's undogmatic atheism. This is why R. J. Berry has noted, "Sadly, debates about evolution and creation tend to divert efforts away from building a robust and refreshed natural theology. This is a tragic legacy of Darwinism" (Berry 2012, p. 38).

Nevertheless, even before Darwin's death in 1882, Wallace had already started natural theology on a dramatically different course—away from Paley's fixist creationism and into a spiritual plurality of evolutionary progress via efficient cause. This would become a new natural theology of a very different order.

## 4. Wallace's Intelligent Evolution

Wallace was never attracted to formal religion of any kind. Unlike Darwin's deistic father and grandfather who imbued Charles with an Enlightenment-infused Whig liberalism that tended toward a quiet religious skepticism, Wallace recalled having Bible stories read to him and growing up on a steady diet of Sunday sermons. Wallace left these religious moorings early on, but he also never embraced the deep-seated materialist reductionism of his more famous colleague. Ever since his youth, experiencing the radical working class politics of the London Mechanic's Institute, Wallace was attracted to the socialistic spiritualism of Robert Owen—what Owen called "the New Moral World"—that regarded all creedal religions as flawed (Barrow 1986, pp. 98, 111–21). While sensitive to the positive aspects of Christ's teachings, Wallace was never a Christian. He could never reconcile himself to a God-made-man atonement, a trinity, or a God of hellfire damnation. But neither was he prone to reducing humans to the highest form of the animal world. He always regarded the difference between *Homo sapiens* and animals as unbridgeable. Even though Wallace would be introduced to the elite circles of Victorian science as the lesser conjoined twin of Darwin, he began distancing himself from the author of *Origin* early on.

That process was launched with his address before the Anthropological Society of London on 1 March 1864: "The Origin of Human Races and the Antiquity of Man Deduced from the 'Theory of Natural Selection'" which would mark Wallace's first great departure from Darwin. Wallace argued that the special "social and sympathetic" qualities arising from humans' mental development set them apart and freed them from the harsh realities of natural selection. The very attributes that mark out human exceptionalism—their cultivation of the land and domestication of animals along with a host of uniquely human mental capacities—freed them from the bondage of natural selection and gave

them mastery over their environments in ways no other species had. Alluding to Richard Owen (famed British anatomist and arch-rival of Darwin), Wallace declared, "we may admit that even those who claim for him a position as an order, a class, or a sub-kingdom by himself, have some reason on their side" (quoted in Flannery 2018, p. 16). When Wallace's address was published, Hooker became alarmed: Might this tacit alliance with Owen suggest an apostasy from the Darwinian camp? At this point, Wallace assuaged Hooker's fears by pointing out his areas of disagreement with Owen and affirming humans as a distinct family within the order of the great apes (Flannery 2018, p. 17). Yet, trouble was brewing.

This distinction was facilitated, in part, by Wallace's close relationship to Charles Lyell, who argued for a naturalistic uniformitarianism in geology, but concluded his *Geological Evidences* (1863) with the assertion that nature "may be the material embodiment of a preconcerted arrangement," and that "the perpetual adaptation of the organic world to new conditions leaves the argument in favour of design, and therefore of a designer, as valid as ever" (quoted in Flannery 2018, p. 21). This signaled the beginning of a common bond with his younger colleague that would become stronger with time. For Lyell, human exceptionalism, especially as demonstrated in their unique intellectual capacities, suggested something more than blind laws operating on matter. As Wallace's biographer, Ross A. Slotten, has observed, "The friendship between Wallace and Lyell was to have a significant impact on Wallace's thinking, especially on his later views of humanity—to Darwin's great distress" (Slotten 2004, p. 218).

The "distress" would be realized by Darwin soon enough. It came in the April 1869 issue of the *Quarterly Review*. In a review of the tenth and latest edition of Lyell's *Principles of Geology*, after long, meticulous, and largely favorable examination of Lyell's work, Wallace made his alliance with Lyell's affirmation of design and deity unmistakable:

> Let us fearlessly admit that the mind of man (itself the living proof of a supreme mind) is able to trace, and to a considerable extent has traced, the laws by means of which the organic no less than the inorganic world has been developed. But let us not shut our eyes to the evidence that an Overruling Intelligence has watched over the action of those laws, so directing variations and so determining their accumulation, as finally to produce an organization sufficiently perfect to admit of, and even to aid in, the indefinite advancement of our mental and moral nature. (quoted in Flannery 2018, p. 53)

What Wallace meant was that "our mental and moral nature" could *not* be explained by blind naturalistic forces alone, and certainly not by natural selection that was for him, as we have seen, an eliminative principle and not a creative one. Of what possible survival advantage could there be in abstract thought, mathematics, music, art, dance, humor, and scores of other attributes universally evident in peoples of all nations and races? He had lived with the most "primitive" of peoples in South America and the Pacific Islands and seen all these attributes on daily display. For Wallace, the answer for them had to come from teleological agents of a spiritual nature. In other words, the mental attributes of *Homo sapiens* must, in turn, be the result of a Mind or a mind-like process. Since Darwin's own theory relied upon this principle of utility, Wallace simply reasoned that this was not a departure from but rather an elaboration upon the Darwin-Wallace theory of evolution. Darwin would show him how wrong he was.

Darwin was appalled. His fears that Wallace had "murdered . . . your own and my child" were now realized, and he thrice underlined the pertinent section in his copy of the *Quarterly* with an emphatic "No!" (Slotten 2004, pp. 268–70). Now Wallace truly had become, in Slotten's words, "a heretic in Darwin's court." Wallace either failed to appreciate how absolutely committed Darwin was to a theory which guided what would be called today 'methodological naturalism'—the idea that the only proper science was that which invoked exclusively natural processes functioning via unbroken natural laws in non-teleological ways—or (more likely) he did not care. For Darwin, this was rank speculation, *not* science; for Wallace, this was where a fair and unfettered view of the evidence led. The two men would remain cordial, but the bond had been broken. Darwin and Wallace continued to correspond,

and they seemed to have a mutual respect for one another; Darwin for Wallace's magnanimity in acknowledging his priority in natural selection, and Wallace for Darwin's opening the door to the elite circles normally closed to a working class "specimen haggler" like Wallace. Despite their differences, an elderly and ailing Darwin even campaigned to get Wallace a government pension, which he did in 1881, at a much-needed £200 per annum.

Wallace continued from that point of departure in 1869 to openly elaborate upon his teleological world. He did so primarily in three publications. First, in chapter 15, "Darwinism Applied to Man," of his book *Darwinism: An Exposition of the Theory of Natural Selection with Some of Its Applications* (1889), Wallace extended teleology from humans alone to three key moments in organic history: First, in the origin of life; second, in the establishment of consciousness or sentience in various degrees and orders of existence now separating plant and animal life; and third, in the endowment of humankind with those "noblest of faculties" that raise us "above the brutes and open up possibilities of almost indefinite advancement." In the first extended elaboration of his teleological theory, Wallace explained: "These three distinct stages of progress from the inorganic world of matter and motion up to man, point clearly to an unseen universe—to a world of spirit, to which the world of matter is altogether subordinate . . . And still more surely can we refer to it those progressive manifestations of Life in the vegetable, the animal, and man—which we may classify as unconscious, conscious, and intellectual life,—and which probably depend upon different degrees of spiritual influx. I have already shown that this involves no necessary infraction of the law of continuity in physical or mental evolution" (quoted in Flannery 2018, p. 193).

What Wallace is suggesting here is that natural laws are efficient causes initiated, guided, and directed by unseen spiritual forces. Thus, there is no transgression of natural laws—no miraculous suspension of them—in the development of what John Herschel called "the mystery of mysteries," the introduction of new organic forms. In effect, Wallace was enunciating *intelligent evolution*, a theory of common descent based on natural selection strictly bounded by the principle of utility (that is, the idea that no organ, attribute, or morphological feature of an organism will be developed and retained unless it affords it a survival advantage) within a larger teleological and theistic context. This is the foundation upon which Wallace built his natural theology. What intelligent evolution most adamantly opposed was the fixist Paleyesque world of what Wallace called "constant interference."

Having established the basic framework of his intelligent evolution in the biological world, Wallace now turned his attention to the cosmos. In *Man's Place in the Universe: A Study of the Results of Scientific Research* (1903), he examined those conditions necessary for the origination and development of life culminating in human beings. These are the conditions he enumerated as 10 intricately balanced and maintained features of the earth absolutely requisite to life: (1) the distance of the planet from the sun; (2) the mass of the earth; (3) the obliquity of its ecliptic orbit; (4) the amount and distribution of water compared to land; (5) the surface distribution of both; (6) the permanence of this distribution related to the moon; (7) an atmosphere of sufficient density with the necessary component gases; (8) adequate dust particles in the air; (9) atmospheric electricity; and (10) a strictly controlled and constantly maintained temperature (Flannery 2018, p. 89). Using these conditions, Wallace mapped out a fine-tuned universe known today as the *anthropic principle*, the idea that these tightly regulated cosmological constants, all pointing to highly endowed human beings, required something beyond mere chance and necessity to explain. This formed the cosmic portion of the teleological argument.

Wallace completed his trilogy of natural theology with his grandest statement of all—his physico-theology—in *The World of Life: A Manifestation of Creative Power, Directive Mind and Ultimate Purpose* (1910). Numerous examples could be given, and, in fact, the entire last half of the book is given over to Wallace's own brand of natural theology expressed simply as, "beyond all the phenomena of nature and their immediate causes and laws there is Mind and Purpose; and that the ultimate purpose is (so far as we can discern) the development of mankind for an enduring spiritual existence" (Wallace 1910, pp. 277–78). For Wallace, between ourselves and a distant, inscrutable "Deity"—a First Cause—there exists an elaborate hierarchy comprised of "grades of beings" with each possessing

successively high powers at development and control (Wallace 1910, p. 393). Wallace had seen Darwin's world, but not as "barren virgins"; rather, as busy and fruitful angelic presences. He inferred the necessity of this kind of direction from the intricate complexity of cell mitosis explicated in detail by August Weismann, from the vitalistic ontogenesis of German physiologist, Max Verworn, and from the "vital force" of plant life and growth presented by Austrian botanist, Anton Kerner.

It is interesting that after presenting some general facts relating to the overall processes of evolution, Wallace launched the natural theology of his system with chapter 13 under the title, "Some Extensions of Darwin's Theory." Because Wallace was convinced that none of his teleological ruminations altered the foundational principles of evolution itself, he insisted that he had not departed from "Darwinism" in the least. Of course, not only did Darwin adamantly disagree but so did Thomas Henry Huxley, Joseph Dalton Hooker, John Tyndall, and many others now wedded not just to a theory of life's development and diversification but to an ideology of science defined by blind laws of nature empirically observed in a closed system demarcated by methodological naturalism. Nonetheless, Wallace was undeterred. For the remaining nearly 150 pages of *The World of Life*, a systematic natural theology is presented in which law-like processes, teleologically orchestrated, replace Paley's fixist interventions. Curiously enough, despite Wallace's derision of Paley's natural world of "constant interference," he was not above utilizing some of Paley's own examples as evidence of design in nature, such as the bird's wing and the feather. Moreover, like Paley's famous watch on the heath example, whereby a designer is deduced from the intricacies of the machinery within the timepiece that make it function properly, Wallace referred repeatedly to "the whole vast complex of the organic machinery" of life requiring design, direction, and purpose (Wallace 1910, pp. 294, 302, 337–38, 360, 395). Wallace's "world of life" was not Paley's world in its execution, but it bore some remarkably Paleyesque characteristics in its analogous attributes.

Well aware that no natural theology could stand without considering the problem of pain and suffering, Wallace constructed his own theodicy in chapter 19, "Is Nature Cruel? The Purpose and Limitations of Pain." Darwin had indicated that he saw too much suffering, pain, and cruelty in nature to suggest any kind of divine presence. Yet, Wallace argued from Darwin's principle of utility that "no organ, no sensation, no faculty arises before it is needed, or in a greater degree than it is needed" such that "we may be sure that all the earlier forms of life possessed the minimum of sensation required for the purposes of their short existence; that anything approaching to what we term 'pain' was unknown to them" (Wallace 1910, pp. 374–75). As for personal loss and pain, Darwin simply gave it over to a blind and indifferent world ruled by "nature red in tooth and claw," while Wallace could see in his own life of privation and suffering the instructive uses for improvement to which such untoward occurrences could be put. For Darwin, the death of his eldest daughter, 10 year-old Annie from tuberculosis in 1851, did much to confirm his loss of faith in a providential God and prompted him to "set the Christian faith firmly behind him" (Keynes 2001, p. 222). For Wallace, the death of his beloved son "Bertie" (nearly seven) of scarlet fever in 1874 brought only scorn for the medical profession and the first installment of his essay, "A Defense of Modern Spiritualism" in the *Fortnightly Review* (Slotten 2004, p. 319). Tragedy meant loss of belief for Darwin; for Wallace, tragedy confirmed belief.

The source for Wallace's own evolution toward natural theology can be found in its very beginning, in his careful delineation of natural versus artificial/human selection. Wallace always thought Darwin had mistakenly conflated the two in his domestic breeding analogy. Again, it must be emphasized that Wallace saw none of his arguments for a teleological and spiritually imbued universe as a departure from the core evolutionary theory he and Darwin had developed. After proposing a "grand law of continuity" controlled by "higher and higher intelligences," Wallace pleaded his case:

> I cannot admit that it in any degree affects the truth or generality of Mr. Darwin's great discovery. It merely shows, that the laws of organic development have been occasionally used for a special end, just as man uses them for his special ends; and, I do not see that the law of "natural selection" can be said to be disproved, if it can be shown that man does not owe his entire physical and mental development to its unaided action, any more than it is

> disproved by the existence of the poodle or the pouter pigeon, the protection of which may
> have been equally beyond its undirected power. (Wallace 1871, p. 370)

Accordingly, there it is—Wallace now used Darwin's own breeding analogy for support of his heretical emendations to evolutionary theory. This difference between natural and artificial/human selection was important to Wallace. Darwin failed to appreciate this distinction and was haunted by the specter of teleology against the hard reality of his chance-based theory.

Thus Wallace could, at least in his own mind, speak as a proponent of "Darwinism" despite the fact that he had become uniquely un-Darwinian. But by the time of his *World of Life*, he had begun to appreciate the degree to which Darwin's theory had become identified with the self-organizing "necessary" conditions of physical laws and properties. He told an interviewer on 4 November 1910, just before the release of his book, "All the errors of those who have distorted the thesis of evolution into something called, inappropriately enough, Darwinism, have arisen from the supposition that life is a consequence of organization. This is unthinkable" (quoted in Flannery 2018, p. 97). Perhaps for Wallace this was true, but clearly not for Darwin.

One final word is in order before leaving Wallace's intelligent evolution. It should not be assumed that even though Wallace could cast his teleological vision within the exemplary framework of Paley that it was merely an extension of it. As Martin Fichman has noted, unlike the natural theology championed by Paley and continued into the nineteenth century through the Bridgewater Treatises, "Wallace was too astute an observer of the vast sociopolitical, environmental, and metaphysical transformations wrought by the (seeming) triumphs of Victorian science and technology to be able to countenance the comforting harmonization of science and religion that had characterized much of eighteenth- and nineteenth-century natural theology. Wallace's mature evolutionary teleology must be viewed as a new response to the challenges posed to questions of human values as science emerged as an increasingly potent and professionalized cultural institution at the start of the twentieth century" (Fichman 2001, pp. 249–50). Fichman is correct, but just as John Polkinghorne, Charles Swinburne, Alvin Plantinga, and others have kept natural theology a viable concern in our own time, Wallace had fashioned a *new* natural theology for a *new* era, one accurately described by Fichman.

This was Paley's thesis in entirely new garb. We should, therefore, not be surprised to find his influences carried forward into the new century as witnessed in physicist, Oliver Lodge; physician/novelist, Arthur Conan Doyle; philosopher, Anthony O'Hear; paleontologist, Robert Broom; neurophysiologist, John C. Eccles; and astrophysicist, Fred Hoyle (Flannery 2018, pp. 69–151 passim). In the end, intelligent evolution has been carried forward into our own generation by intelligent design (ID) proponents, and none so powerfully and persuasively as biochemist Michael Behe. Behe acknowledges Paley's sometimes awkward and ill-chosen examples, and his tangential theological asides, but he regards Paley's watchmaker argument as scientifically valid. Adam Shapiro has carefully traced the evolving meanings of Paley's natural theology through time, concluding that Behe and many of his ID colleagues through court, press, and public presentation, "tacitly won the right to proclaim themselves Paley's true inheritors, and by extension assert their own standing as the legitimate foils to Darwin" (Shapiro 2014, p. 122).

Although Shapiro is clearly alluding to Behe's testimony in the famous (or infamous) *Kitzmiller v. Dover* case in 2005, Behe remains today no less emphatic in his support of Paley's use of "sophisticated work in biology" *and* his acknowledgment of Wallace's early embrace of intelligent design (Behe 2019, pp. 4–5). "Although one has to take care in constructing a valid design argument and Paley admittedly overreaches in places in his watchmaker argument," writes Behe, "his main point is exactly correct: *we recognize design in the purposeful arrangement of parts* [emphasis in the original]" (Behe 2019, p. 88). Wallace does nothing less. Wallace's complaint against Paley is his fixist and "constant interference" approach, not his design argument per se.

Shapiro may be broadly correct, but it must be remembered that the natural theology of Wallace, whose intelligent evolution fits so nicely into Behe's current framework, was not entirely Paley's; especially if we refer back to the definition of natural theology that introduced this discussion,

"the idea that there exists some link between the world we observe and another transcendent realm." If Wallace—and by implication, Behe—is merely trading in religion, one is at a loss for defining what kind. Even Fichman admits that "Wallace's theism resists simplistic classification" (Fichman 2001, p. 247). Wallace sought spiritual regeneration of what he regarded as a rapacious mammon-driven capitalist industrialism, and he felt that most institutional religions were too bound up with that corrupt political economy to be committed—much less passionate—instruments of regenerative reform.

Thus, Wallace fits in neatly with a broad-brushed definition of natural theology emphasizing connections between transcendental realms. The question of transcendency is a philosophical and metaphysical one with major religious implications, but it could hardly be called religious since it fails to define a single specific precept, principle, or doctrine relative to it. Wallace's was a natural theology without a church, mosque, or temple. After all, natural theology can only take us so far; it is not the *revealed* theology or sacred wisdom of the Torah, Bible, Qur'an, or Buddhavacana. Neither was it a Paleyan natural theology aimed at revealing the wonders of a Christian God, showing the benevolent care "on the part of the artist [God], a constant attention to this property of his work, distinct from every other" forming "what most properly ought to be accounted the blessings of Providence" (Paley 1802, pp. 132, 345).

## 5. Finding Wallace in the Historical Fog

The story might end there, but unfortunately, the idea that Wallace refurbished natural theology to be compatible with modern evolutionary theory is not widely recognized. Of the sizeable Wallace historiography, too long to list here, only historian Martin Fichman and I have followed completely down this path.[4] It is fair to say that Wallace's metaphysical views have been wrapped in a historical fog of three different kinds: First, those who refuse to see or acknowledge Wallace's theistic beliefs; second, a view that tends to diminish and make superfluous Wallace's natural theology by arguing for a fully compatible Darwinian theism; and third, a variety of misconstructions of Wallace based upon poor historical analysis and argumentation. In attempting to clear away these layers of fog, it is not my goal to argue for acceptance of Wallace's natural theology. Wallace's speculations may or may not be sound inferences; they stand or fall on metaphysical grounds well beyond the scope of this paper. This is not meant to be a Wallace apologetic. Its goal is far more modest; it is simply to clear away the fog of denial, misinterpretation, and historical fallacy that has enveloped the Wallace historiography.

This first group that denies Wallace's theism almost defies reply. It is akin to arguing the age of the earth with someone who has decided it is 6000 years old; no facts are going to intrude on this faith-filled conviction. Surprisingly, this denial comes from librarian/biogeographer at Western Kentucky University, Charles H. Smith, an acknowledged authority on Wallace. Smith likes to concentrate on certain Wallace writings and ignore or "reinterpret" others. I have dealt with Smith's views in some detail in *Nature's Prophet* and will not repeat it here (Flannery 2018, pp. 124–26). Similarly, is the suggestion by Tim Flannery that Wallace offered a proto-Gaia hypothesis shorn of theistic intent, or worse, Steven J. Dick's insistence that Wallace was a "self-proclaimed atheist" (Flannery 2018, pp. 123–24). To all these confused fog-makers I simply suggest a straightforward reading of Wallace's trilogy of natural theology or of his personal correspondence. For example, late in life, he confided to his friend and biographer James Marchant, "The completely materialistic mind of my youth and early manhood has been slowly moulded into the socialistic, spiritualistic, and theistic mind I now exhibit" (Marchant 1916, vol. 2, p. 181). This definitive proof against any rendering of a non-theistic Wallace has long been published and readily available; to ignore it seems inexcusable.

---

4   I have cited Fichman's paper, "Science in Theistic Contexts," because he elaborates on the way in which Wallace's scientific theism introduced a new way of addressing the sociopolitical and cultural landscape of the Victorian/Edwardian eras and fitted itself into nexus of science and religion in the emergent twentieth century. More broad-ranging and highly recommended is the full book-length treatment available in Fichman's *An Elusive Victorian: The Evolution of Alfred Russel Wallace* (2004).

Yet, the second layer of fog is a bit more formidable. It argues that Darwin's evolutionary theory can be regarded as fully compatible with theism. Biologist Ken Miller, physicist Karl Giberson, and NIH director Francis Collins have all made such arguments. This Darwinian theism has its own issues that I have addressed elsewhere (Flannery 2017). But more germane to our interests here are the assertions of Cambridge professor Alister McGrath, who, like Stephen Dilley, uses "the laws impressed on matter by the Creator" passage in *Origin* to confirm a form of deism in Darwin. Only McGrath is more emphatic, insisting, "there is not a whiff of personal atheism here. It is difficult to believe that his references to a Creator in the *Origin of Species* were simply contrived to mollify his audience, representing crude deceptions aimed at masking a private atheism that Darwin feared might discredit his theory in the eyes of the religious public" (McGrath 2011, pp. 159–60). The fact that McGrath can articulate this view so well suggests that perhaps it is harder to dismiss than the superficial asseveration of incredulity he offers against it. As we have seen earlier, an examination of Darwin's notebooks suggests a materialism incompatible with theism. Furthermore, in Darwin's private correspondence with men like Asa Gray and Charles Lyell, it is clear that he opposed any real theistic interpretation to his evolutionary theory. "Darwin's principle of natural selection was chosen by him precisely *because* it excluded any creative action by God. That is why he was so upset with Lyell and Wallace and murmured against Gray. They kept letting God in. We should not be fooled," writes Benjamin Wiker, "by his [Darwin's] sop about a Creator added to the *Origin*" (Wiker 2009, p. 139).

If the abovementioned is not enough, Darwin's meeting with the two leading atheists of the day, Ludwig Büchner and Edward Aveling, toward the end of his life is revealing. Joining the three was Darwin's son, Francis. During that meeting, Darwin is said to have argued, "I am with you [Büchner and Aveling] in thought, but I should prefer the word Agnostic to Atheist." For Darwin, "'Agnostic' was but 'Atheist' writ respectable, and 'Atheist' was only 'Agnostic' writ aggressive." Then Darwin asked, "Why should you be so aggressive? Is anything gained by trying to force these new ideas upon the mass of mankind? It is all very well for educated, cultured, thoughtful people; but are the masses yet ripe for it" (Aveling 1883, p. 5). Thus, for Darwin, agnosticism translated into an operational undogmatic atheism. All that God-talk was rhetorical window dressing aimed at softening the blow of a godless universe for his reading public. Interestingly, Francis's edited version of his father's *Autobiography* attempted to distance his father from these atheists. Francis argued that readers might be "misled" into thinking his views were closer to Büchner's and Aveling's than they might appear. However, even he had to admit that Aveling's summary of their meeting "gives quite fairly his impressions of my father's views" (Darwin [1893] 2000, p. 69).

McGrath is convinced "that a renewed natural theology is indeed capable of engaging with a Darwinian view of reality. Teleology, for example, may have been redefined; it has not been destroyed or invalidated" (McGrath 2011, p. 268).[5] But it is hard to envision a natural theology meaningfully "engaged" with a theory intentionally designed to rule it out. "Here Darwin," writes David Kohn, "the last of the natural theologians, is the man who turned out the lights" (Kohn 1989, p. 238). Darwin's God-talk was his eulogy to a dead concept. Yet, the lights Darwin turned off, Wallace turned back on.

Beyond the fog of denial—namely, denying Wallace's theism—and the fog of affirming Darwinian theism, hovers the third, and perhaps most daunting fog of all, that of fallacious reasoning and poor historical argumentation. Here, I apply the premise established long ago by David Hackett Fischer "that there is a tacit logic of historical thought" and it is expressed as "a process of *adductive* reasoning in the simple sense of adducing answers to specific questions, so that a satisfactory explanatory 'fit' is obtained" (Fischer 1970, p. xv).

The abovementioned is a more general kind of mistake in failing to distinguish Wallace's particular use of natural selection within a context of natural theology from Darwin's reductionist use. It has been convincingly demonstrated by Adam Shapiro that there are clear connections between

---

5      On definitions of teleology, see the accompanying essay in this issue.

Wallace's intelligent evolution, its current ID proponents, and even Paley himself. Shapiro's analysis notwithstanding, it is surprising to find in a book devoted to natural theology no attempt to distinguish between Wallace's and Darwin's views (De Cruz and De Smedt 2015). Instead, Wallace and Darwin are joined together by Helen De Cruz and Johan De Smedt as naturalists "seriously weakening" the design argument contrasted with complaints of ID proponents "misrepresenting or altogether ignoring" natural selection, despite the fact that Wallace along with most ID proponents devote reams of paper to discussing this principle (De Cruz and De Smedt 2015, p. 79). Here are two historians' fallacies combined: the fallacy of the hasty if not insidious generalization and the fallacy of presumptive continuity (Fischer 1970, pp. 124–25, 154–55). In the first, generalizations are presented as plain facts with no support or confirmatory evidence; in the second, the ideas of Wallace and Darwin are conjoined as the originators of natural selection with no recognition of the fact that the two evolutionists came to very different conclusions; conclusions drawn from details in their respective theories that were very different from the very beginning.

Something quite similar is committed by Boyd Barnes. Accepting Smith's earlier work, Barnes insists that "Wallace should not be understood to have defected from Darwinism in favor of spiritualism but to have been consistent in his larger understanding of natural selection and evolutionary theory" (Barnes 2008, p. 200). This is right and wrong. With regard to Wallace's spiritualism, Smith and Barnes are right in not seeing spiritualism as the main catalyst to his defection from the Darwinian camp. As we have seen, Wallace had separated humanity from the tyranny of natural selection over a year before he experienced his first spiritualist phenomena on 22 July 1865. In this sense, spiritualism merely added to the importance of human exceptionalism that formed a foundational concept in all of Wallace's work. Additionally, Wallace's idiosyncratic position within the development of modern evolutionary theory allowed him to see his immersion into spiritualism as much of an extension of Darwinism as his teleology. Wallace regarded spiritualism not as mysticism but as science, a "new branch" of anthropology, as he called it. Further, Efram Sera-Shriar writes "Because it combined both familiar aspects of Darwinian evolution with monogenesis and progressivism, his theory of spiritualism could be incorporated into larger ethnological and anthropological discussions about human evolution. However, it moved beyond these discussions into new ground, and it provided an alternative framework that accounted not only for the evolution of the living, but also for the dead" (Sera-Shriar 2020, p. 371). This, then, became woven into the fabric of Wallace's comprehensive natural theology. Rather than seeing spiritualism as a causative factor (real or imagined) in Wallace's views, it seems more accurate to simply incorporate spiritualism into Wallace's broader scientific approach that rejected methodological naturalism in favor of an extended role for logical inference and the evidentiary power of "credible witnessing" (Sera-Shriar 2020).

It is harder to maintain, as Smith and Barnes do, Wallace's allegiance to Darwin's essential theory. It must be remembered that Wallace's insistence that he was merely "extending" Darwinism was *his* belief, certainly not Darwin's or any of his close allies. Barnes' discussion of Darwin and Wallace in terms of natural theology are largely undifferentiated, therefore, this becomes problematic (Barnes 2008, pp. 200–17). Barnes believes Wallace belongs "as part of the broader category of Darwinian scientific thought" (Barnes 2008, p. 217). But, as we have seen, Wallace's persistent disagreement with Darwin over his human/artificial selection analogy reveals a deep-seated difference in the functional operations of natural selection, not to mention the fact that Wallace always considered natural selection a principle of species elimination rather than creation. In linking Darwin and Wallace so closely together, Barnes has also committed the fallacy of presumptive continuity.

John O. Reiss has presented a more sophisticated and nuanced discussion of Darwin and Wallace in relation to natural selection. Reiss speaks of three different kinds of teleology (Reiss 2009, pp. 9–17). There is teleological determinism (goal-directed final causes from known features of the environment and organism); external representational teleology (ideas in the mind of God); and internal representational teleology (goal-directed vital principles). These terms, however, seem rather clumsy and imprecise. For example, teleological determinism is more akin to teleonomy (things that appear

purposeful only in the sense that the organism does something and adapts itself to conditions "in order to" survive and/or reproduce); external representational teleology is best associated with that of William Paley and the Bridgewater Treatises; and internal representational teleology may be associated with orthogenesis, some prominent proponents of which include Lamarck, Teilhard de Chardin and Henri Bergson. The blurry boundaries of Reiss's categories can be seen in Wallace, who best fits within both external *and* internal representational teleology. Reiss believes that Wallace had fewer issues with teleology than Darwin because of his closer connection to the principle conditions of existence, but this supposedly came at the cost of a further separation from the principles of inheritance. I doubt that this argument holds together, mainly because most of Darwin's assertions regarding inheritance—his adoption of pangenesis—were wrong, and Wallace's fewer problems with teleology stem not from his views on the conditions for existence but from his adoption of an intrinsically teleological worldview.

According to Reiss, "The paradox of Darwin's perspective is that he wanted to produce a naturalistic theory of evolution but began by accepting the premise of the design argument, which is fundamentally antinaturalistic" (Reiss 2009, p. 148). He does so by claiming that Darwin used natural selection as a replacement for direct design. In other words, Darwin "considered selection to have solved the design problem Paley and Lyell posed for him" (Reiss 2009, p. 133). But in what sense does this mean that Darwin "accepted" the design premise? Reiss suffers from an equivocation in the usage of teleology.[6] By his own definition, teleological determinism would never have been solely acceptable to Paley. Reiss, at the same time, claims that Wallace avoided teleology in his original formulation of his version of natural selection but fails to address Wallace's increasing acceptance of external *and* internal representational teleology (to adopt his cumbersome terminology) in his intelligent evolution. If Darwin felt a certain tension between design and teleology on the one hand, and evolution and methodological naturalism on the other, he is right for the wrong reason. It was not his Paleyan heritage that haunted him, but rather the expectations of his readers that vexed him.

We find this same sort of equivocation in David Kohn. Kohn is, I believe, quite correct in identifying at least part of Darwin's "pulling of his metaphysical punches" to the influence of his religious fiancé and wife, Emma (Kohn 1989, p. 226). But holding back on one's metaphysical opinions cannot explain Darwin's ambiguity in these matters. According to Kohn, "Darwin's new teleology was to go beyond dropping final cause as an 'anomaly' of language. Natural selection brought with it a critical restructuring of natural theology. Simultaneous to the jettisoning of providentialist final cause came a new understanding of harmony in nature as the product of competitive struggle" (Kohn 1989, p. 229). But is this teleology anymore? Is this a "critical restructuring of natural theology" or a jettisoning of the concept altogether? Can Darwin really be regarded as "the last of the natural theologians"? It would appear these assertions are the products, once again, of an equivocation of now *two* terms, *teleology* and *natural theology*. As it stands, this fallacy of equivocation lends to more confusion than clarity (Fischer 1970, pp. 274–75).

Tim Lewens has more recently ramped up the fog machine with his discussion of "neo-Paleyan biology," defined as giving "Paley credit for having identified the central problem for the natural historian, while adding that he entirely mistook the true process that answers that problem" (Lewens 2019). According to Lewens, neo-Paleyans—this includes famed atheist Richard Dawkins—acknowledge the legitimate search for design in nature, but argue that natural selection provides the answers to this search. They praise Paley for understanding that mere chance alone could never serve as a mechanism for such "exquisite adaptations," instead finding its source not in the "mysticism" (Lewens' term) of intelligent design but in Darwin's principle of natural selection. While it is true that, technically speaking, natural selection is not a random process, it is placed within a broader context of factors such as genetic mutation, random drift, and environmental change that clearly *are*. Darwin did not know about genetics, of course, but he was convinced that Paley was utterly mistaken

---

6    This sort of equivocation is sorted out in the accompanying essay in this issue.

and that there was "no more design in the variability of organic beings, and in the action of natural selection, than in the course which the wind blows" (Darwin [1958] 2005, p. 73). The problem here, again, is one of equivocation on what exactly is *Paleyan*. Would Paley himself accept the attribution of his name to such a formulation as Lewens proposes? Here, Lewens—and those who self-identify as *neo-Paleyans*—shift the meaning of Paley's natural theology into one of biological reductionism. Paley surely did not believe he was engaging in mysticism when he spoke of an intelligent designer. Here, the meaning of a term is altered as it changes hands, leading to argumentative distortion (Fischer 1970, pp. 275–76).

Finally, there is Harvard evolutionary biologist and historian, Andrew Berry, who dismisses my argument that Wallace not only developed a refurbished natural theology but that it was essentially compatible with orthodox Christianity (Flannery 2018, pp. 116–17).[7] Berry emphatically counters, "There is nothing remotely Christian about Wallace's teleology. In fact, he carefully disavows Christianity, insisting in his conclusion to *The World of Life* that postulating 'the Infinite and Eternal Being as the one and only direct agent in every detail of the universe seems, to me, absurd'; his is the spiritualist's universe, entailing 'infinite grades of influence of higher beings upon lower'" (Berry 2019, p. 859).

Nevertheless, Berry is giving Wallace a careless and superficial reading. Wallace is *not* making an argument against God or a First Cause, only one that acts as a "*direct agent in every detail of the universe.*" Instead, this First Cause or ultimate Mind *entails* a spiritual universe of "infinite grades of influence of higher beings upon lower." The absence of a detectable First Cause did not mean it was nonexistent, only that it was mediated by other spiritual agents (Flannery 2011, p. 36). Wallace is clear about this: "I can imagine the supreme, the Infinite being, foreseeing and determining the broad outlines of a universe which would, in due course and with efficient guidance, produce the required result" (Wallace 1910, p. 393). This was no deistic front-loading.

The Anglican priest, John Magens Mello, immediately understood that Wallace's *World of Life* spoke to God's angelic beings acting upon and directing the affairs of humankind and the natural world that was familiar to Christians, and he wrote an extensive summary and review praising it.[8] Furthermore, Aiden Nichols, a Dominican priest, associated Wallace's metaphysic to Aquinas's idea "that God governs inferior things through superior ones. The First Cause gives being; secondary causes determine it" (quoted in Flannery 2018, p. 72).

For Wallace, a first cause could never be detected in the natural world because all empirically-based causes are efficient ones activated and directed by subordinate spiritual beings that Christians typically call angels. The idea of an angelic hierarchy dates back at least to the Syrian monk Pseudo-Dionysius the Areopagite (circa fifth century) who introduced and inserted the whole of pagan Neo-Platonism into a Christian context. Berry's superficial reading of Wallace is matched only by his apparent unfamiliarity with Christian doctrine and history. The fallacy of misplaced precision seems apparent here (Fischer 1970, pp. 61–62). Because Wallace does not specifically and explicitly endorse Christianity, Berry assumes his metaphysic cannot possibly accommodate it. Berry simply reads Wallace's words without interpretation or context. This fallacy of factual verification completely ignores a rich Christian

---

7  It may seem contradictory to admit that Wallace was not a Christian while his natural theology was *compatible* with orthodox Christianity, or for that matter, with all the Abrahamic religions. In order to understand how this can be, the limited explanatory capacity of *natural* theology must be distinguished from the more robust claims of *revealed* theology. All Wallace ever said in his natural theology is that there is an inscrutable "eternal Mind" governing a law-like universe that is managed through subordinate "hosts of angels" or "organizing spirits." He said no more and no less in his trilogy of works. From this perspective, "the limited nature of natural theology is not necessarily a theological problem. Let us assume that natural theology offers a proof for the existence of God, who has properties x, y, and z [e.g., immutability, immateriality, and dominion]. Trinitarian-revealed theology argues for the existence of God, who has properties x, y, z, b, and d. Does natural theology offer a twisted account of God? I do not think so. Just like I can acquire knowledge about Bruce Wayne without knowing that he is also Batman, it should be possible to use natural reason to acquire knowledge about the God of theism—without the doctrine of Trinity" (Vainio 2017, p. 15). The minimalism of natural theology has within it the ability to find a non-Christian presenting a quite traditional Christian view of a Creator and His creation precisely *because* no doctrinal issue is touched.

8  This is elaborated upon in his essay "The Mystery of Life and Mind" reprinted in its entirety with a biographical note on Mello (Flannery 2011, pp. 219–41).

history broadly encompassed in Wallace's trilogy of works—human exceptionalism, humanity's cosmic significance, and an overarching hierarchy of angelic orders arranged to carry out a divine plan. More astonishingly, Berry fails to see the relevance of Wallace's construction of a human-centered cosmology that reflects much of scripture. Historically speaking, Berry can surely disagree with it but he is hardly at liberty to ignore its relevance. If this is, in fact, true—and everything in Wallace's trilogy of natural selection suggests it—it goes precisely to the point of Wallace's compatibility with Christianity. Berry cannot argue for a position or interpretation, namely, that there is "nothing remotely Christian about Wallace's teleology," while at the same time summarily dismissing evidence that plainly contradicts that very argument. This is mere hand-waving.

## 6. Conclusions

In clearing away the historical fog that has hidden and obscured Wallace as a meaningful figure in the history of ideas generally and the history of natural theology specifically, we find that the fog of denial is contradicted by a plain reading of Wallace's primary resources; the argument for a Darwinian theism that would make Wallace' natural theology superfluous rests on a weak historical foundation; and the remainder of this tortured historiography has been plagued with assorted examples of faulty adductive reasoning—fallacies of generalization, narration, semantical distortion, and factual verification are all in evidence. Once cleared away Wallace stands as the new progeny of Paley's natural theology, now revised and transformed to accommodate common descent by means of natural selection with a theodicy shorn of its Pollyannaish conceits. Paley may have been Darwin's foil, but Wallace can stand within a more robust tradition of natural theology that has withstood David Hume's skepticism, Immanuel Kant's epistemic presumptions, and even Karl Barth's faith-precedes-reason brand of extreme fideism. Thoughtful analysts all agree, natural theology is alive and well (Polkinghorne 2006; Swinburne 2007; Vainio 2017). Hopefully now, Wallace's historiography is as well.

Wallace's trilogy of natural theology was not written to prove God. John Polkinghorne reminds us that the "new natural theology" makes no apologetic claims. It *does* demonstrate that "theistic belief is not . . . logically inevitable, but that it gives us the deepest and most satisfying *insight* into the way the world is. It is not that our atheistic friends are stupid—far from it—but that atheism explains less than theism can" (Polkinghorne 2006, p. 171). Wallace would have agreed.

**Funding:** This research received no external funding.

**Conflicts of Interest:** The author declares no conflict of interest.

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
