# Peer review of "Alfred Russel Wallace’s Intelligent Evolution and Natural Theology"

_religions, doi:10.3390/rel11060316_

Round 1

Reviewer 1 Report

This is an excellent article, clearly written, persuasively argued, and well documented.  It gives an excellent analysis of the history and nature of natural theology, and the place of Darwin and Wallace in that tradition.  I found it to be highly illuminating.  I would strongly recommend publishing it as it is, with only typographical errors corrected.

Author Response

Thank you for your supportive review. Rest assured that I will thoroughly proof the ms. before publication.

Reviewer 2 Report

I thought it’s an insightful and important treatment of Wallace, and fair when often he is overlooked or dismissed. It’s original and shows a real command of the historical moves needed to unpack his work. Enjoyed it and learned from it. 

Author Response

Thank you for your supportive review. Rest assured I will thoroughly proof the ms. before publication.

Reviewer 3 Report

Alfred Russel Wallace’s Intelligent Evolution and Natural Theology

This article is a fine exposition of the differing views of Alfred Russel Wallace and Charles Darwin on the ramifications of the theory of evolution by natural selection, for which they are both generally given intellectual credit. The author of this article argues persuasively against other writers on the topic that, although their proposed mechanism of evolution was superficially similar, Wallace and Darwin disagreed fundamentally about its scope and about what ultimately was driving it. For example, Wallace thought that the intellectual abilities of humanity were beyond evolution by natural selection, and he argued his position based exactly on the Darwinian “principle of utility.” Thus, while Darwin’s view of evolution tended strongly to materialism, Wallace’s view led to a new natural theology — one that went beyond the excessively cheerful thinking of William Paley. This paper is an important contribution to understanding both Wallace and the ways in which evolution may be understood.

I noticed a few typographical errors:

line 37: there should be a semi-colon after “theology”
line 59: “eighteen-century” should be “eighteenth-century”
line 432" “August Weimann” should be “August Weismann”

Author Response

Thank you for your supportive review. I also appreciate your noting the following typos, which I will correct immediately as well as make a complete proof of the ms, before publication.